# Properties and Performance Verification on Magnetite Polydimethylsiloxane Graphene Array Microwave Sensor

**DOI:** 10.3390/polym13193254

**Published:** 2021-09-24

**Authors:** Mohd Aminudin Jamlos, Mohd Faizal Jamlos, Azri Alias, Mohamad Shaiful Abdul Karim, Wan Azani Mustafa, Prayoot Akkaraekthalin

**Affiliations:** 1Faculty of Electronic Engineering Technology, Universiti Malaysia Perlis, Kampus Alam UniMAP, Pauh Putra, Arau 02600, Malaysia; wanazani@unimap.edu.my; 2College of Engineering, Universiti Malaysia Pahang, Kuantan 26300, Malaysia; mshaiful@ump.edu.my; 3Faculty of Mechanical and Automotive Engineering Technology, College of Engineering Technology, Universiti Malaysia Pahang, Pekan 26600, Malaysia; azribalias@ump.edu.my; 4Faculty of Engineering, King Mongkut’s University of Technology North Bangkok, Bangkok 10800, Thailand

**Keywords:** PDMS graphene array sensor, UWB spectrum, PDMS Ferrite

## Abstract

This paper investigates the use of a Magnetite Polydimethylsiloxane (PDMS) Graphene array sensor in ultra-wide band (UWB) spectrum for microwave imaging applications operated within 4.0–8.0 GHz. The proposed array microwave sensor comprises a Graphene array radiating patch, as well as ground and transmission lines with a substrate of Magnetite PDMS-Ferrite, which is fed by 50 Ω coaxial ports. The Magnetite PDMS substrate associated with low permittivity and low loss tangent realized bandwidth enhancement and the high conductivity of graphene, contributing to a high gain of the UWB array antenna. The combination of 30% (ferrite) and 70% (PDMS) as the sensor’s substrate resulted in low permittivity as well as a low loss tangent of 2.6 and 0.01, respectively. The sensor radiated within the UWB band frequency of 2.2–11.2 (GHz) with great energy emitted in the range of 3.5–15.7 dB. Maximum energy of 15.7 dB with 90 × 45 (mm) in small size realized the integration of the sensor for a microwave detection system. The material components of sensor could be implemented for solar panel.

## 1. Introduction

Nowadays, the application of antennas as sensors is stressed most in terms of bandwidth and efficiency improvement despite miniaturization [1]. Substrate materials could play a vital role in influencing the characteristics stated above [2]. Normally, fer and Rogers as well as the FR-4 substrate are utilized to fabricate the antenna sensors for the conventional method. Reducing the wavelength of the relative permittivity dielectric guide could improve sensor bandwidth and efficiency, as well as the miniaturization of the sensor dimension [3,4]. However, these types of techniques result in thin bandwidth [5], low efficiency, and isolation [6], as well as reduce impedance matching [7].

Recently, a sensor made of a polymer composite substrate with preferable mechanical and electrical characteristics has attracted the interest of researchers. Various kinds of elements are composed with a variety of additions, possibly nickel ferrite, ceramic, and titanium [8]. One of the significant gains of applying dielectric with magnetic based relative permeability as well as permittivity to exceed the unity (εr and μr > 1) [9] is the miniaturization of the size of the probe [10]. Furthermore, such substrates also able to enhance the operated bandwidth of the sensor mentioned in [11,12]. 

Physically, sensors for microwave imaging must be compact, light, and planar [13]. Another study emphasizes that a sensor to detect a tumor must have features of low profile, high resolution (higher bandwidth), less complexity, lightweight, good efficiency, and good radiation directivity [14]. Other techniques for improving operated bandwidth that require additional structures towards the sensor, such as metasurface, metamaterial, and planar inverted, are not suitable to be applied and integrated into the sensor due to bulky and uneven structures which lead to uneven permeability and permittivity [15,16]. Polymeric substrates, such as the magnetic element of ferrite, which is well-distributed in nanoscale, have contributed to the stability of permittivity and permeability. Hence, it is very essential to have bandwidth improvement without compromising the other ideal features of the sensor, such as light, compact, low complexity, and low profile, which can be achieved by improvising the polymerization of the sensor’s substrate [15].

PDMS is one the kind associated with flexibility, less permittivity, resistance to water, as well as the ability of mixing other elements to enable a variety of fabricated substrates for the antenna [17,18,19]. Incorporating PDMS and the magnetic element of ferrite could realize sensor bandwidth enhancement, as explained by Hansem and Burke, with the thickness (*t*) effect of a magneto-dielectric towards bandwidth enhancement [12], as demonstrated in Equation (1):(1)$$BW≅\;\frac{96\cdot \sqrt{\frac{\mu_{r}}{\varepsilon_{r}}\;\;\frac{t}{\lambda_{0}}}}{\sqrt{2\;}\cdot [4+17\cdot \sqrt{\mu_{r}\;\varepsilon_{r}}]}$$

On the other hand, Graphene has gained much interest among researchers to replace copper as a radiating material of the patch antenna sensor. A comprehensive Graphene study revealed graphene to be totally advanced compared to copper, single and multi-wall carbon nanotubes [20,21]. The features include 3.3 × 10^−1^ nm (nano size), 10^8^ s/m (high conductivity), 7.7 × 10^−1^ mg/1 m (light element), and 150 × 10^6^ psi (strong element) of the single layer carbon atoms in 2-D honeycomb lattice constructed the Graphene. 

As for detection purposes, using a microwave frequency range called microwave imaging, bandwidth and gain of the sensor are among the most important elements to be considered compared with others element [22]. The antenna performance relates to bandwidth, gain, radiation pattern directionality, and antenna efficiency, and these are essential to collect scattered signals to form a high-resolution image. Apart from that, the vital properties of graphene, namely high conductivity which leads to high gain, will serve as a promising alternative sensor material for cancer detection purposes due to the higher microwave signal required for human head structure penetration, including skin and skull, in order for the signal to reach the brain [23].

In this paper, a Graphene sensor array, which is fully embedded in a composite substrate of magnetic-based PDMS dielectric (PDMS-Ferrite), is presented as well as the study of the combination efficiency of the substrate and the graphene. This is done by comparing the proposed antenna sensor with a similarly dimensioned antenna fabricated on a dielectric (Taconic) substrate. Although the concept of PDMS-magneto materials has been reported in literature recently, the validation study towards the scientific relation between polymer dielectric with magnetic-based substrates and a Graphene sensor in terms of the bandwidth and gain described in this paper is among the earliest.

An overview of such antennas published in the literature is presented in Table 1 to determine that the antenna proposed in this work results in a significant bandwidth enhancement and gain in comparison to the state-of-the art. The low permittivity and low loss tangent of PDMS-Ferrite as well as the high conductivity of graphene have significantly contributed to the wide bandwidth and high gain of the Magnetite PDMS Graphene array sensor. Such gain, bandwidth, and small dimensions of 90 × 45 mm^2^ make the UWB sensor suitable for microwave detection purposes [24].

## 2. Sensor Design and Fabrication Method

The magnetite PDMS Graphene array sensor is generally constructed of Graphene to form an array circular patch, incorporate a feeding line, parasitic element, and partial ground, as well as PDMS-Ferrite to form magnetite dielectric substrate. For simulation, Graphene has been set to 3.8 × 10^8^ s/m for the conductivity value and PDMS-Ferrite is set to 2.6 and 0.01 for permittivity and loss tangent respectively. The array circular patch has been measured through specific calculation prior to being simulated using computer simulation software. Once being improved, the optimized patch size was determined at 15 mm in diameter. A 50 Ω microstrip line acts as feed source placed at the center right back of the patch for direct connection between the feed source and patch. In order to realize high gain sensor, a single circular patch is multiplied to produce a 4 × 1 array. A corporate feeding structure is applied by adding feedlines for the connectivity of each patch. Identical power delivery of all patches could be realized by applying a quarter-wave transformer impedance matching method. Hence, 100 Ω and 50 Ω feedlines are coordinated using quarter wave transformers of 70.71 Ω [30].

Figure 1 demonstrates the front, back, and top dimension of the proposed simulated sensor, constructed by graphene patches with the thickness of 0.03 nm and PDMS-Ferrite for the novel substrate element with thickness of 1.6 mm. The introduction of a parasitic element as the new technique to realize ultra-wide band has the width of 8 mm and length of 32 mm with 0.02 gap with the transmission line. The parasitic element is the conductor material located near to radiating elements, yet it is not electrically connected with them. It functioned to provide electromagnetic coupling with the radiating elements by acting as a passive resonator. The waves from the parasitic elements do interfere by strengthening the antenna’s radiation in the desired direction and cancelling out the waves in undesired directions [31]. The details of the geographical sizes of the sensor are demonstrated in Table 2.

Figure 2 explains in detail the processing steps (flowchart) for synthesizing the Magnetite PDMS Graphene array sensor. The PDMS used has the permittivity of 2.8 while the graphene used has the carbon content, thickness, and conductivity of 97%, 25 µm, and 3.5 × 10^8^ s/m, respectively [32]. On the other hand, Figure 3 demonstrates the synthesized Magnetite PDMS Graphene array sensor.

## 3. Result and Discussion

Magnetic Ferrite in three different ratios (25%, 30%, and 35%) and PDMS (75%, 70%, and 65%) is well blended respectively to realize three different mixed solutions of novel PDMS-Ferrite dielectric substrate with magnetic characteristics and a diversity of unique features. It is proven that the ratio of 30% iron oxide nanoparticles leads to improving dielectric properties which directly enhance the sensor performance. However, the sensor performance started to degrade once the ratio achieved 50% of Ferrite because of the excessive magnetic element existence leads to high magnetic losses. Figure 4 exhibits permittivity and loss tangent values according to different mixture composition of PDMS-Ferrite. Figure 4 demonstrates that 30% of ferrite concentration ratios recorded the lowest permittivity (2.6) and nearest value of zero (0.01) loss tangent constantly across the operated frequency as compared with concentration of 25% and 35%. Hence, 30% and 70% of ferrite and PDMS concentration, respectively, is the ideal ratio to form the novel substrates for having a larger, more useful bandwidth. The negative loss tangent of the PDMS (75%) and Ferrite (25%) mixture, as shown in Figure 4b, is mainly obtained by the metallicity of the nanocomposites and the plasma oscillation of delocalized electrons. Theoretically, the negative permittivity can be explained by the Drude model. Due to the mass of atomic nucleus being much more than the electrons, the dipole phenomenon is attributed to the movement of a large number of electrons [33].

The PDMS-Ferrite substrate has been measured in terms of permittivity as well as loss tangent using a dielectric probe and VNA. The PDMS-Ferrite substrate measurement results for permittivity as well as loss tangent with 30% of Ferrite concentration in the form of liquid and solid are shown in Figure 5. Averages of 2.7 and 0.01 are the values recorded for permittivity and loss tangent measurement results respectively for the liquid PDMS-Ferrite substrate. Nevertheless, these values slightly increased to averages of 2.8 and 0.047 once PDMS-Ferrite substrate turned into solid form. On the other hand, the PDMS-Ferrite substrate recorded 1.4 for permeability determined using the Nicolson–Ross–Weir approach, as explained by [33]. Due to the different element construct FR4 and the proposed substrate, which are fiberglass and PDMS-Ferrite respectively, the sensor with FR4 substrate would exhibit a small bandwidth as compared to a sensor with PDMS-Ferrite as substrate which demonstrates large bandwidth because of the correlated effects of permeability, permittivity, and loss tangent [33].

On the other hand, the waveguide approach is another method to determine PDMS-Ferrite permeability. Figure 6a illustrates how the approach applies the strong electric field at the waveguide center while Figure 6b shows the strong magnetic field at the waveguide wall. Hence, material permittivity and permeability will be dominant as the material located at the middle and at the side wall respectively during transmission coefficient (S21). Figure 6c shows the measurement setup, which consists of an adapter for the coaxial waveguide, network analyzer, and G-band waveguide (support frequency: 3.95−5.85 GHz), while Figure 6d indicates the location of sample for testing within the waveguide. Nevertheless, we failed to obtain a permeability calculation due to the extremely small reading of the sidewall, which resulted in an error through the transmission coefficient (S21). Thus, PDMS-Ferrite substrate electrical properties did not significantly alter with the presence of 30% magnetic Ferrite. Due to that, the determination of the permittivity value obtained by taking into account the permeability of the sample is the air permeability obtained through the fitting method [34].

A study has been performed to investigate the optimum thickness of PDMS-Ferrite by varying the thickness of simulated PDMS-Ferrite in the range of 0.4–2.0 mm, as shown in Figure 7. Based on the reflection coefficient (S11) simulated results shown in Figure 8, the optimum thickness of PDMS-Ferrite substrate is 1.6 mm, as proposed in this paper due to the widest operated frequency (9 GHz) recorded, from 2.2 GHz to 11.2 GHz, as compared with the other operated frequency thicknesses. 

The FESEM image of graphene in Figure 9a illustrates the formation of scattered groups of graphene sheets with a clear sign stacking. The graphene sheets look like a layer sheet morphology more than the outsized area owing to the large surface to volume ratio. A similar morphological nature of the graphene as closely stacked due to the elimination of an oxygen group to form a closely associated stack arrangement [35]. Meanwhile, XRD patterns of graphene are presented in Figure 9b, where the most intensive characteristic peak observed at 26.60° corresponds to the (002) plane. This is clearly indicating the formation of pure graphite. As presented in Figure 9, the stacking sheets are closely linked to each other due to the interplanar distance and d-spacing reduction. The existence of graphene single layer structures and vanishing multilayer component will increase the greater spreading peak. The decreasing of the size of sheets with a shapeless arrangement is formed from a broad hump at 2θ = 20–30° [36].

A four point probe device was used to measure graphene conductivity, as shown in Figure 10. From the measurements, graphene is found to feature a conductivity of 3.38 × 10^8^ s/m, resistivity of 2.01 × 10^−3^ Ω.m, voltage of 0.2 V, and current of 9.48 × 10^−1^ A. The measured high conductivity and low resistivity values are good in agreement with early reported work on the conductivity and resistivity of graphene by several researchers [37]. The observed electrical properties are enough to be utilized as a conductive material to replace copper and other metals for various applications. The chemical with microwave assisted combination process-based reduction method is playing a prime role to enhance the electrical properties [38].

Fabricated sensors of the Magnetite PDMS Graphene array and Copper-Taconic are illustrated in Figure 11a,b correspondingly while Figure 12a,b presents the measured comparisons of S-parameters and gain of the two sensors. The proposed Magnetite PDMS Graphene array sensor operated within the frequency range of 2.2–11.2 (GHz) while Copper-Taconic resonates within the range of 4.2–11.2 (GHz). The gain of Magnetite PDMS Graphene array sensor is significantly higher (15.7 dB) than Copper-Taconic, which is 13.5 dB. The bandwidth and maximum gain of the Magnetite PDMS Graphene array sensor has been increased up to 28% (2 GHz) and 2.2 dB respectively. The ideal dielectric properties of low permittivity and low loss tangent of Magnetite PDMS-Ferrite as well as the high conductivity of graphene significantly contributed to the wide bandwidth and high gain of the Magnetite PDMS Graphene array sensor. 

On the other hand, the Magnetite PDMS Graphene array sensor E-plane radiation pattern at frequency of 3, 4, 5, and 6 GHz is shown in Figure 13. Those frequencies achieved among the highest gain and the longest wavelengths recorded by the sensor. A uni-directional radiation pattern recorded ensured that the sensor was able to emit the signals throughout ultra-wideband frequency [39]. The patterns exhibit the average directivity of 7.4 dB while the highest was recorded at 15.7 dB. 

## 4. Conclusions

In a nutshell, the addition of a magnetic element, Ferrite, to a PDMS solution enables to reduce the permittivity value and loss tangent of the sensor substrate. Ferrite and PDMS with in the proportion of 30% and 70% each were selected and integrated to realize low permittivity as well as low loss tangent of 2.6 and 0.01 correspondingly for the Magnetite PDMS Graphene array sensor substrate. The proposed sensor recorded UWB operated frequency within the range of 2.2−11.2 (GHz) and great energy was emitted in the range of 3.5–15.7 dB due to ferrite relative permeability as well as permittivity exceeding the unity (εr and μr > 1) and low permittivity of PDMS. The sensor with PDMS-Ferrite substrate managed to have bandwidth enlargement as much as 28%, which is equal to 2 GHz for this paper as compared with the conventional Taconic substrate. In terms of the gain, the proposed Magnetite PDMS Graphene Array sensor recorded the maximum gain of 15.7 dB compared with Copper-Taconic antenna which recorded 13.5 dB for maximum gain due to the presence of graphene as the radiating element that has a conductivity value (3.7 × 10^8^) higher than copper (3.7 × 10^7^).

## Figures and Tables

**Figure 1 polymers-13-03254-f001:**
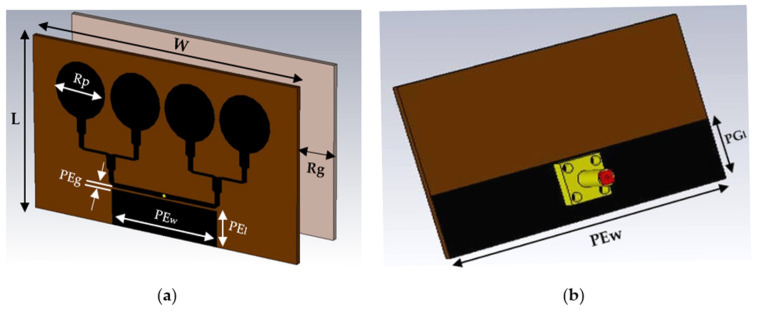
Simulated graphical propose sensor, (**a**) front and (**b**) back (transparent without reflector).

**Figure 2 polymers-13-03254-f002:**
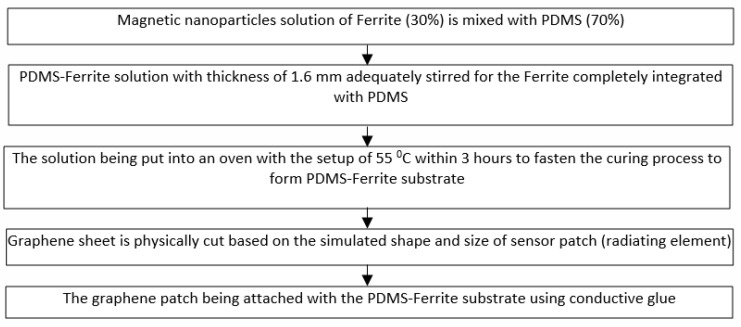
Processing steps for synthesizing the Magnetite PDMS Graphene Array Sensor.

**Figure 3 polymers-13-03254-f003:**
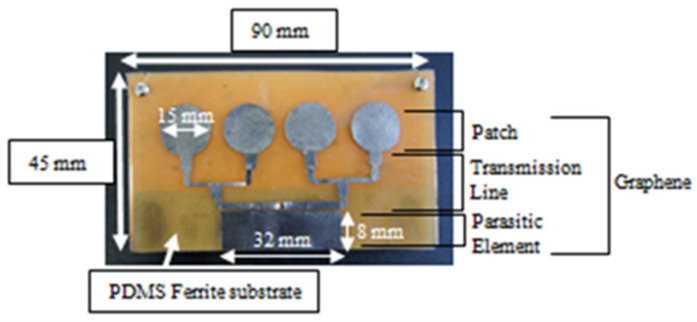
The synthesized Magnetite PDMS Graphene Array Sensor.

**Figure 4 polymers-13-03254-f004:**
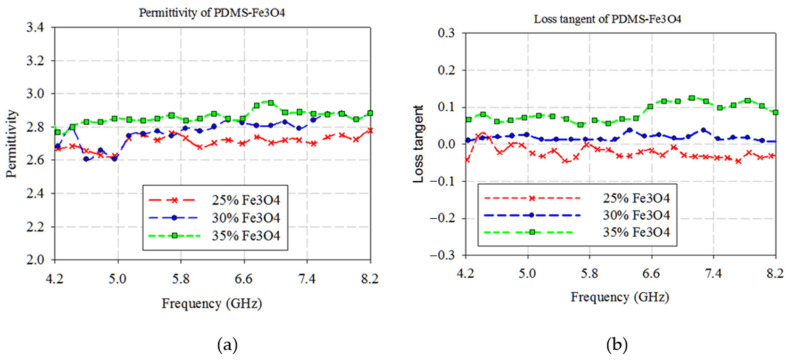
PDMS−Ferrite measurement results for permittivity (**a**) and loss tangent (**b**).

**Figure 5 polymers-13-03254-f005:**
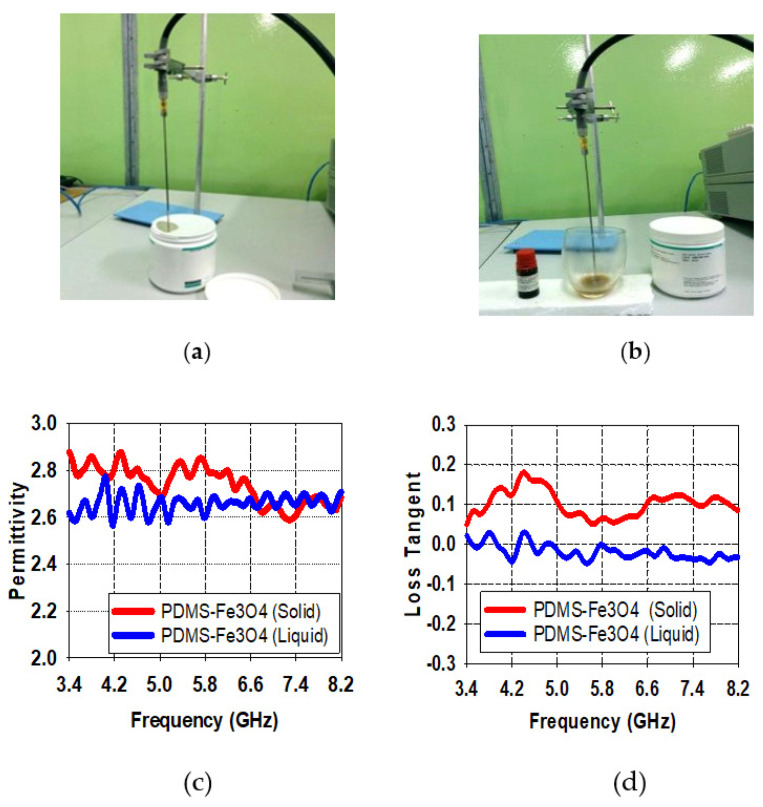
PDMS-Ferrite measurement results for dielectric characteristic; (**a**) liquid state and (**b**) solid state, result of measured permittivity PDMS in liquid and solid cases (**c**), and result of measured loss tangent PDMS in liquid and solid cases (**d**).

**Figure 6 polymers-13-03254-f006:**
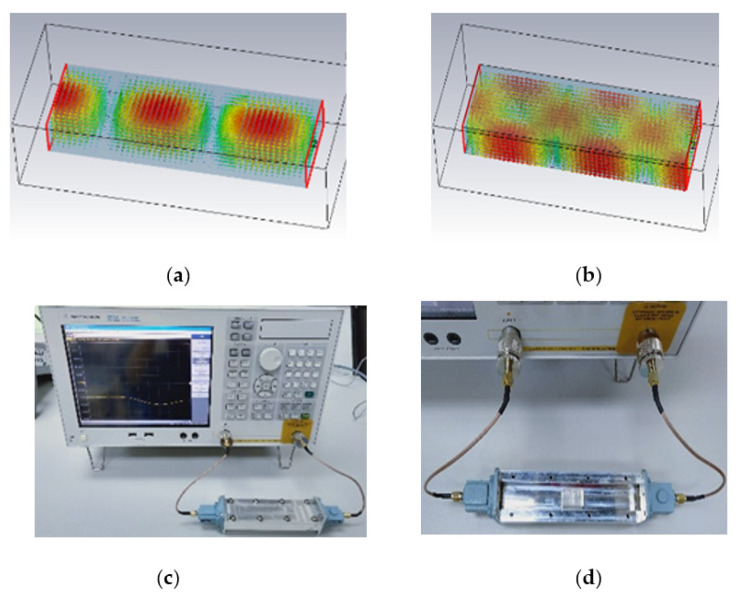
Waveguide technique for solid PDMS-Ferrite dielectric characteristic measurement; (**a**) strong electric filed at the waveguide center, (**b**) strong magnetic field at the waveguide wall, (**c**) measurement setup using waveguide technique, and (**d**) sample location for testing within the waveguide.

**Figure 7 polymers-13-03254-f007:**
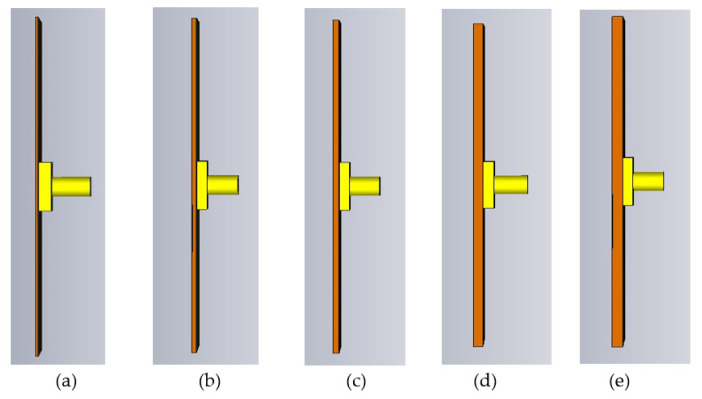
PDMS-Ferrite substrate thickness; (**a**) 0.4 mm, (**b**) 0.8 mm, (**c**) 1.2 mm, (**d**) 1.6 mm and (**e**) 2.0 mm.

**Figure 8 polymers-13-03254-f008:**
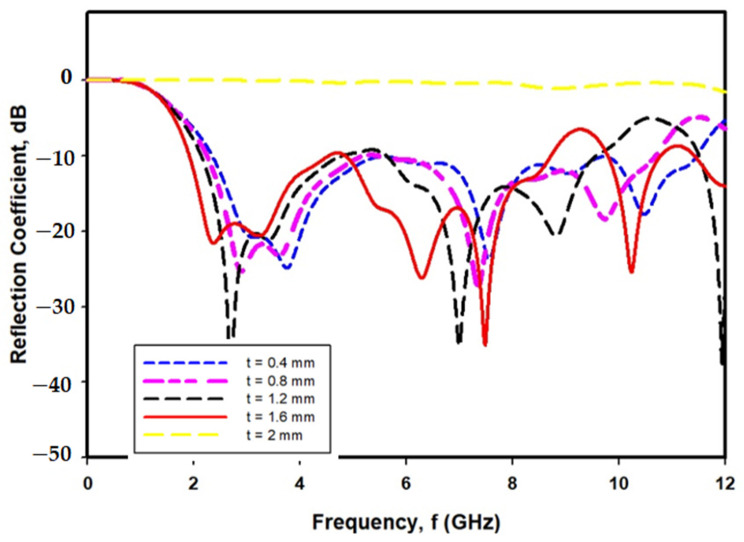
Operated bandwidth for Magnetite−PDMS Graphene Array sensor with different thickness of substrate.

**Figure 9 polymers-13-03254-f009:**
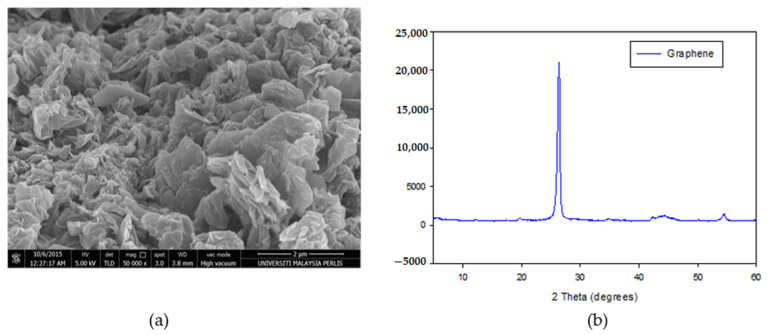
Characterization of Graphene (**a**) FESEM images (50,000 times of magnifying), (**b**) XRD pattern.

**Figure 10 polymers-13-03254-f010:**
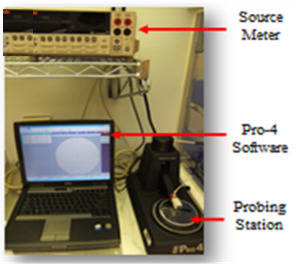
Four point probe device.

**Figure 11 polymers-13-03254-f011:**
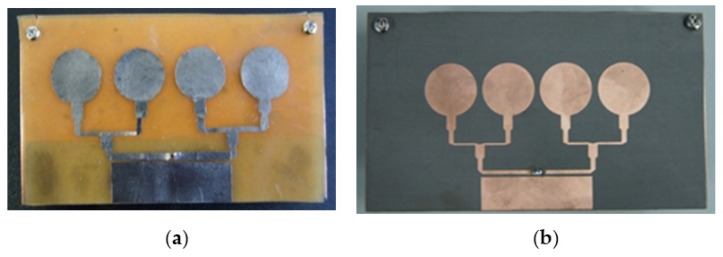
Comparison of fabricated antenna sensors (**a**) Magnetite PDMS Graphene Arraysensor and (**b**) Copper-Taconic sensor.

**Figure 12 polymers-13-03254-f012:**
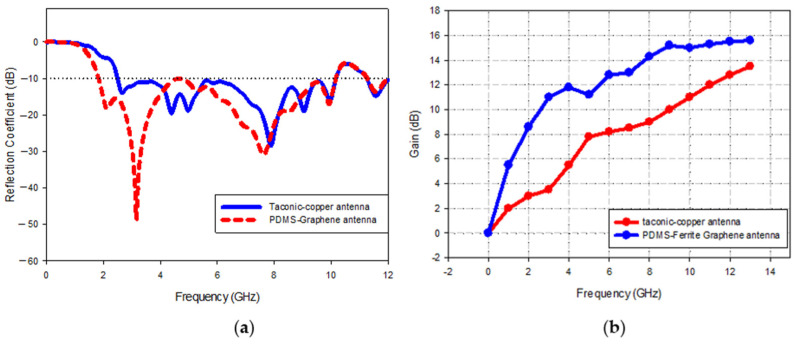
Magnetite PDMS Graphene Array and Copper−Taconic Sensor; (**a**) operated bandwidth, (**b**) gain.

**Figure 13 polymers-13-03254-f013:**
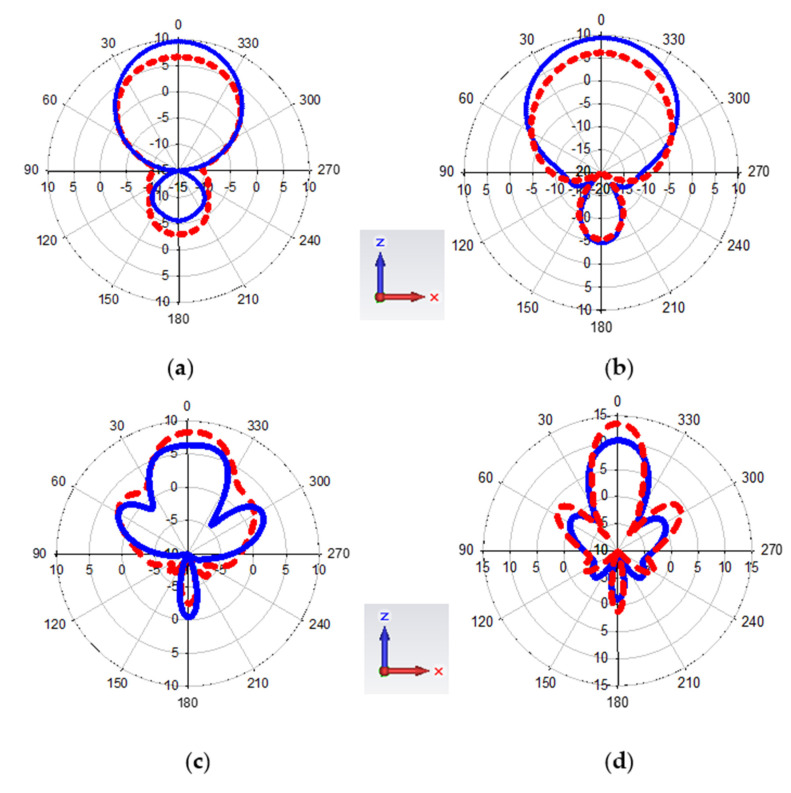
E-plane radiation pattern in red colour (simulated) and blue colour (measured) for (**a**) 3 GHz, (**b**) 4 GHz, (**c**) 5 GHz, (**d**) 6 GHz.

**Table 1 polymers-13-03254-t001:** Different Antennas with Magneto-Dielectric Substrates.

Ref	Center Frequency (GHz)	Bandwidth (GHz)	Max Gain (dB)
[25]	2.1	0.039	2.4
[26]	0.85	0.1	5
[27]	0.8	0.077	5.1
[28]	2	0.2	1.5
[29]	2.0	6	5.3
This work (Magnetite PDMS Graphene Array sensor	4.9	9	15.7

**Table 2 polymers-13-03254-t002:** Geometrical Dimensions.

Parameter	Value (mm)
*W*	90
*L*	45
*Rp*	15
*PE* * _g_ *	0.2
*PE* * _l_ *	8
*PE* * _w_ *	30
*PG* * _l_ *	15
*PG* * _w_ *	32
*R* * _g_ *	20

## Data Availability

Not applicable.

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
