# Peer review of "Properties and Performance Verification on Magnetite Polydimethylsiloxane Graphene Array Microwave Sensor"

_polymers, 2021, doi:10.3390/polym13193254_

Round 1
Reviewer 1 Report
Reviewer report on Manuscript Draft entitled ‘Properties and Performance Verification on Polydimethylsiloxane Magnetite Graphene Array Microwave Sensor’.
The aim of this work is to investigate the application of magnetite Polydimethylsiloxane (PDMS) Graphene array sensor in Ultra-Wide band (UWB) spectrum for microwave imaging application. The proposed array microwave sensor comprises of Graphene array radiating patch, ground and transmission lines with substrate of Magnetite PDMS-Ferrite.
Significant corrections were performed after previous assessment round.
Presented research and discussions are valuable and interesting from the point of view of practical application of composite polymers. The research is in the scope of the journal. Therefore, the manuscript eventually can be published after some minor improvements and corrections:
Title and Abstract: Title is not very clearly formulated, it should be improved, where is necessary comas should be added and other punctuation marks should be applied. The same should be done in these parts of Abstract where array structure is addressed.
Introduction: The application of graphene-based structures in various research directions is mentioned in Introduction, but not strongly supported by references. Therefore, a reference (Single-Walled Carbon Nanotube Based Coating Modified with Reduced Graphene Oxide for the Design of Amperometric Biosensors. Materials Science & Engineering C 2019, 98, 515–523) is recommended to support applicability of graphene-based structures.
Experimental part contains too many basic photo images, that should be either removed either replaced into supplementary material if authors think that they are valuable for readers.
Results and discussion: Figure 4, photo images for this figure are not very informative, could be either removed either replaced into supplementary material if authors think that they are valuable for readers.
Author Response
Answer to Reviewer for Properties and Performance Verification on Polydimethylsiloxane Magnetite Graphene Array Microwave Sensor
Reviewer 1
please see the attachment.

Reviewer 2 Report
In this manuscript, magnetite PDMS graphene array-enhanced sensor was developed and manufactured to operate along a wide range of radio frequency spectrum. Using both numerical studies and experimental investigations, it is shown that the designed low-loss device utilizes the both bandwidth enhancement and high conductivity of the atomically thin graphene sheet to realize high gain at ultrawideband frequencies. Ultimately, the assessed the detection capabilities of the designed sensing tool through experimental verifications. Although the claims in the work supported with both simulation and experiments, it suffers from important shortcomings. I listed my remarks below and the authors have to address them carefully in the revision through conducting comprehensive modifications.
1) The writing quality of the manuscript must be enhanced carefully. There are several poor sentences and expressions along the manuscript, which reduce the quality of the work. For instance, "as far as the authors know" does not make sense at all. All similar expressions must be avoided. On the other hand, "...there is no publication yet on validation study towards the scientific relation between polymer dielectric with magnetic based substrates and Graphene sensor in term of the bandwidth and gain" must be rewritten in more professional way. Here also, all analogous sentences must be avoided and corrected.
2) The Introduction section of the work is too weak and the authors have to discuss the recent state of the art approaches in the field of RF antennas and metastructures in the bibliography section.
3) Most of the panels in Figure 2a-2f are not required and do not provide any scientific knowledge. Hence, those panels must be removed.
4) What is the limit of the detection of the designed sensor? This parameter in highly critical in assessing the performance and quality of a sensor. For more details see: ACS Photonics 3, 2308−2314 (2016), Materials Today 32, 108-130 (2020).
5) In addition to the comment above, the sensitivity and figure of merit (FoM) are the other key parameters of a sensor that must be quantified.
6) Although the design and development of high-gain and efficient antennas is significantly important, they do not play critical role compared to mode volume and Q-factor of the excited states. This must be compared and argued.
7) It must be demonstrated that how the system operates at different doping levels of the graphene monolayer? It is not clear that if the graphene layer was in intrinsic phase or other states. This needs to be clarified.
Round 2
Reviewer 2 Report
In the revised version of the manuscript, non of the recommended and requested comments have been responded. Therefore, I cannot recommend the work for publication in this format. I will ask the authors to conduct anothr round of revisions.
Author Response
Dear reviewer,
Please see the attachment.
Sincerely,
Faizal

Round 3
Reviewer 2 Report
Publishable.